# Haplotype Purging after Relaxation of Selection in Lines of Chickens That Had Undergone Long-Term Selection for High and Low Body Weight

**DOI:** 10.3390/genes11060630

**Published:** 2020-06-08

**Authors:** Yunzhou Yang, Yanjun Zan, Christa F. Honaker, Paul B. Siegel, Örjan Carlborg

**Affiliations:** 1Institute of Animal Husbandry & Veterinary Science, Shanghai Academy of Agricultural Sciences, Shanghai 201106, China; yang.yunzhou@imbim.uu.se; 2Department of Medical Biochemistry and Microbiology, Uppsala University, 75123 Uppsala, Sweden; yanjunzan@gmail.com; 3Department of Animal and Poultry Sciences, Virginia Polytechnic Institute and State University, Blacksburg, VA 24061, USA; chonaker@vt.edu (C.F.H.); pbsiegel@vt.edu (P.B.S.)

**Keywords:** Virginia chicken lines, directional selection, relaxed selection, purging, haplotype, domestication, advanced intercross line, body weight

## Abstract

Bi-directional selection for increased and decreased 56-day body weights (BW56) has been applied to two lines of White Plymouth Rock chickens—the Virginia high (HWS) and low (LWS) body weight lines. Correlated responses have been observed, including negative effects on traits related to fitness. Here, we use high and low body weight as proxies for fitness. On a genome-wide level, relaxed lines (HWR, LWR) bred from HWS and LWS purged some genetic variants in the selected lines. Whole-genome re-sequencing was here used to identify individual loci where alleles that accumulated during directional selection were purged when selection was relaxed. In total, 11 loci with significant purging signals were identified, five in the low (LW) and six in the high (HW) body weight lineages. Associations between purged haplotypes in these loci and BW56 were tested in an advanced intercross line (AIL). Two loci with purging signals and haplotype associations to BW56 are particularly interesting for further functional characterization, one locus on chromosome 6 in the LW covering the sour-taste receptor gene *PKD2L1*, a functional candidate gene for the decreased appetite observed in the LWS and a locus on chromosome 20 in the HW containing a skeletal muscle hypertrophy gene, *DNTTIP1*.

## 1. Introduction

The domestication of plants and animals has, for thousands of years, played a pivotal role in the development of human societies [1]. During domestication, imposed artificial selection has been used to change traits of interest. This systematic selection has, for example, resulted in improvements in production traits, making domesticated species, such as chickens, central for supporting the nutritional requirements of humans throughout the world [2]. Domestication involves changes whereby the breeding and care of plants and animals are directly or indirectly controlled by humans [3] and is a continuing process that may be viewed historically in a geographical and time context [1]. The chicken of today crosses freely with its junglefowl ancestor and produces progeny that reproduce [4,5]. The recent (in the context of time) rediscovery of Mendelism and technological developments have resulted in the worldwide development of chickens bred specifically for egg production or body size. This delineation has occurred because of the negative genetic correlation between them, resulting in competition in the allocation of resources [6]. Experimentally selected populations are powerful resources to study the genetics underlying long-term direct and correlated selection responses observed during domestication. For example, studies of populations under directional selection have provided insights to the individual loci involved [7]. When relaxing selection on the directional selected populations, natural selection will come into play. By comparing the genomes of directionally selected lines, and relaxed sublines from them, the purging of variants accumulating under directional selection can be quantified [8]. Since purging is expected for loci involved in negatively correlated selection responses, this approach can also be used to identify candidate loci for further studies into the genetic basis for this effect. Little is, however, still known about the individual loci involved in purging in domestic animals and more research on this topic is therefore valuable as a complement to other more classical mapping approaches.

Here, we have studied an experimental chicken population to identify candidate loci involved in negatively correlated selection responses for later inclusion in more detailed functional genetic studies. These populations, the Virginia body weight lines, originated from founders that were crosses of seven moderately inbred lines of White Plymouth Rocks. Today, it consists of two populations that have been divergently selected for a single trait, either increased (High Weight Selected line; HWS) or decreased (Low Weight Selected line; LWS) 56-day body weight.

Based on changes in haplotype-frequencies after relaxation of selection in the Virginia body weight chicken lines, we aimed to identify individual purging loci and the haplotypes segregating in them using whole-genome re-sequencing and 60 K chicken single nucleotide polymorphism (SNP) chip genotype data from selected (HWS and LWS) and relaxed (HWR and LWR) lines. Fitness is difficult to define for an individual chicken, and even more difficult to measure. Here we used high and low 56-day body weights as proxies for it. This was done under the assumption that haplotypes accumulating in the selected lines, due to their contribution to the artificially imposed fitness criterion (body weight being the only selected trait), are expected to be purged upon relaxation of selection if they also have negative correlated effects on, for example, reproduction or health. Such effects of the identified purged haplotypes on body weight were also evaluated in a large, independent advanced intercross line, generated by crossing chickens from the HWS and LWS lines [9,10]. Using this approach, the objective of the study was to identify individual purging loci in both the HW and LW lineages and to highlight several of these as likely contributors to negatively correlated selection responses in the HWS and LWS for subsequent fine-mapping and functional characterization.

## 2. Materials and Methods 

### 2.1. Ethics Statement

All procedures involving animals used in this experiment were carried out in accordance with the Virginia Tech Animal Care and Use Committee protocols #15-144 and #18-151.

### 2.2. Animals and Phenotyping

The Virginia body weight lines were initiated in 1957 from a founder population that was a cross between seven partially inbred lines of White Plymouth Rocks chickens. The average 56-day body weights of the founders were 878 g in males and 708 g in females (Table 1). In each generation (one generation per year), male and female chickens were selected for high or low 56-day body weight [11]. Numbers of sires and dams selected to produce each line were 8 and 48, respectively, through the fourth generation of selection (XWS4), 12 and 48 from XWS5 to XWS25, 14 and 56 after XWS25. These parents were selected from within-line groups of approximately 150 to 250 individuals, depending on the generation. All chickens from one generation were hatched on the same day of the year, with the same dietary formulation for all generations. In generation S41, chickens from HWS and LWS were crossed to produce an advanced intercross line (AIL) [9,10]. In generation S44, new relaxed lines (HWR and LWR) were established for HWS and LWS where selection on body weight was discontinued to allow for natural selection. For the relaxed lines, random mating was implemented where pooled semen collected from all males (*n* = 10) was used to inseminate females in that generation (*n* = 30) [9,10,12,13]. The chickens used in this study were from (i) generations S41, 50 and 53 of the HWS (*n*_41_ = 29; *n*_50_ = 49; *n*_53_ = 20) and LWS (*n*_41_ = 30; *n*_50_ = 10; *n*_53_ = 20) [10,14], (ii) generation 9 of the relaxed selection lines corresponding to generation S53 of the Virginia high (HWR; *n* = 20) and low (LWR; *n* = 19) body weight selected lines [12], and (iii) chickens (*n* = 2667) from intercross generations F_2_–F_15_ of the AIL [8,13,14,15,16]. (Table 1; Appendix A). Body weights at 56 days of age were measured for the advanced intercross chickens from generation F_2_ to F_15_.

Raw sequencing data, phenotypes and scripts were deposited in sequence read archive (SRA) database (PRJNA631642) and https://github.com/tuzixuexi/Purging-Analysis, respectively.

### 2.3. Genotyping

#### 2.3.1. Genotype Imputation Using Whole-Genome Re-Sequencing and Single Nucleotide Polymorphism Chip Genotype Data

For chickens from generations S41, S50 and S53 of HWS/LWS and generation R9 HWR/LWR, 60K chicken SNP-chip genotypes were available from an earlier study [8,14]. Also, individual whole-genome re-sequencing (WGS) data on HWS/LWS chickens from generation S41 (~30X) were available from an earlier study [17]. Here, *BEAGLE4* was used to impute genotypes from SNP-chip to WGS SNP density in the chickens from HWS/LWS generations 50 and 53, and the HWR/LWR generation 9 [18]. Before imputation, the WGS data were filtered using the following criteria: (1) minor allele frequency (MAF) > 0.05; (2) only bi-allelic SNPs; (3) genotyping quality (GQ) = 20 & mapping quality (MQ) = 50; (4) no genotype missing rate. This resulted in 5,524,212 remaining SNPs on the 28 largest autosomes. From the SNP-chip genotypes, only the 29,147 SNPs that were genotyped also in the WGS datasets were kept, whereas the remaining 24,166 markers were filtered out (Appendix A). After imputation, a custom Python script (https://github.com/tuzixuexi/Purging-Analysis) was used to filter imputed genotypes to only keep those markers with an estimated genotyping probability (GP) above 0.9 in at least 90% of the samples. The dataset after this filtering contained 3,051,963 high quality imputed SNP-genotypes. Before screening for selection signatures in the high and low body weight lineages, a second filtering was performed to only retain SNPs with minor allele frequencies across all generations greater than 0.05, leaving 2,182,770 and 1,348,038 SNPs in the LW and HW lineages, respectively.

#### 2.3.2. Low Coverage Sequencing (LCS) and Genotype Estimation in the Advanced Intercross Line

For 2982 individuals from AIL generations F_1_ to F_15_, whole-genome re-sequencing libraries were prepared using a low-cost *Tn5*-based protocol. Sequencing was then performed to ~0.4X coverage (mean = 0.48; range 0.003–3.2; *n* = 16 individuals with < 400 reads removed) on either an Illumina HiSeq4000 or HiSeq X10 [17,19]. Genotypes were imputed for the individual samples using *STITCH* [20] against the Gallus_gallus-5.0 reference genome (GCA_000002315.3) [21]. To select the *K*-value in imputations, 10 individuals from generation F_0_ were down-sampled from ~30X to ~0.4X coverage using *seqtk* (https://github.com/lh3/seqtk). They were used as control samples to calculate imputation concordance between genotypes imputed from LCS by *STITCH* with different *K*-values and genotypes called from the deep-coverage sequence data on the same individuals. Analyses with different values of *K* were performed for GGA28 (RefSeq: NC_006115.4), and based on them, a *K*-value of 10 and number of generations to 19 were selected, because they provided an acceptable balance between time consumed in the analyses and the resulting imputation quality (Appendix A).

### 2.4. Identifying Purging Loci and Exploring Their Effects on Body Weight

Haplotypes containing alleles that decrease/increase body-weight increase in frequency during selection in the LW/HW lineages, respectively. The loci containing these haplotypes can be detected by doing selective sweep analysis among chickens from subsequent generations [8,14]. The frequencies of selected variants that have correlated negative effects on fitness traits are likely to change slower than for alleles without such effects and consequently remain segregating in the selected lines for more generations. When directional selection for body weight was discontinued in the relaxed lines, it was expected that haplotypes with undesirable correlated effects would be purged from the population by natural selection. These purging loci can thus be detected as selective sweeps in genome-wide comparisons of chickens from selected and relaxed lines. Here, a three-step process was used to first identify individual purging regions and then to evaluate the effects of haplotypes segregating within these lines. This to ultimately identify loci with alleles that contribute negatively correlated effects on body weight and fitness traits. First, selective sweep analyses were used to detect loci under selection [22] in the LW/HW lineages. Purging loci were identified as selected loci where sweeps were due to changes in frequency between selected and relaxed lines. Next, haplotypes were defined and their frequencies estimated in the selected and relaxed lines. Finally, the effects of the segregating haplotypes on body weight were estimated in the AIL generated from crosses of chickens from HWS and LWS in generation S41. Loci most likely to contribute to negatively correlated selection responses were defined as those where purged haplotypes had opposite effects on the selected trait (56-day body weight) to those in the selected lines.

#### 2.4.1. Mapping Regions with Rapidly Changing Haplotype Frequencies

A genome-wide mapping of regions displaying rapid haplotype-frequency changes after relaxation of selection were performed in the LW (LWS41/50/53; LWR9) and HW (HWS41/50/53; HWR9) lineages separately using *hapFLK* [22]. The ancestral haplotype value (K) in these analyses was set to 10. This number was chosen because we assumed that there would be few segregating ancestral haplotypes in generation S41 due to (i) the Virginia lines being founded from seven partially inbred lines [11] and (ii) because they had been subjected to long-term single-trait selection. The significance (*p*-values) for the *hapFLK* results were computed [22]. The *R stats* package was used to identify candidate purging loci at a 5% FDR significance level. The significance thresholds for the LW/HW analyses were 10^−3.3^/10^−3.2^, respectively.

#### 2.4.2. Identifying Purging Loci

To distinguish purging regions from regions undergoing directional selection or drift, *hapFLK* population trees were built and plotted both for the genome-wide genotypic data and the putative selected regions [22]. For ongoing selection, a hierarchical relationship among the analyzed populations of (XWS41, (XWR9(XWS50, XWS53))) (X = L or H) was expected, because the relaxed lines originated from the selected lines at generation 43. Purging regions were defined as those with hierarchical relationships of (XWR9, (XWS41(XWS50, XWS53))) i.e., where the relaxed line becomes an outgroup due to opposite selection on the haplotypes than in the selected lines.

#### 2.4.3. Testing for Body Weight Associations in Purging Regions

Multiple regression modeling was used to test for associations between haplotypes in the candidate purging regions and 56-day body weight in the AIL, which was from generation 43 LWS and HWS chickens. Sex and generation were included in the regression model as fixed effects. For each of the purging regions, a set of markers was first selected based on having large divergence in allele-frequencies between selected (LWS/HWS) and the corresponding relaxed (LWR/HWR) lines, using the criteria of an allele frequency difference (AFD) above 0.5 between the group of selected (generations S41, S50 & S53) and the relaxed line (Appendix A). As a first step to explore 56-day body weight associations in the purging regions, a backward-elimination (BE) analysis, using a 5% False Discovery Rate (FDR) termination criteria [16,23], was used to identify SNPs in each locus that were independently associated with 56-day body weight. It was implemented to, in each candidate purging region, perform analyses linearly across the region, including 200 markers at a time based on their orders on chromosomes. Several analyses were therefore performed when there were > 200 markers in the region. The linear model used in these analyses was:y_i_ = μ + g_i_ + s_i_ + m_i1_ +…+ m_in_ + e_i_,(1)
where *y_i_*, *g_i_*, *s_i_* were the 56-day body weight, generation, and sex of the i^th^ individual, *m_i1_… m_in_* were the genotypes of i^th^ individual for the *n* SNPs to be tested (coded as 0,1,2), and *e_i_* was the normally distributed residual error.

From the markers selected in the backward-elimination analysis, haplotypes were defined for each candidate region using *BEAGLE4* [18]. For each major haplotype segregating in the evaluated lineage (frequency > 0.1), an association analysis was performed using the following one-way analysis of variance (ANOVA) model:y_i_ = μ + g_i_ + s_i_ + h_i_ + e_i_,(2)
where *y_i_*, *g_i_*, *s_i_*and **e_i_** were the same as described in Equation (1), *h_i_* was the number of copies of the tested haplotype (coded as 0,1,2) carried by i^th^ individual.

## 3. Results

### 3.1. Mapping Regions with Changing Haplotype Frequencies within the Low and High Body Weight Lineages

Using *hapFLK*, multiple regions with significant haplotype frequency changes were mapped in the LW and HW lineages (Figure 1). In the LW lineage, significant signals (FDR < 5%) were detected in 24 regions on chromosomes 1, 2, 3, 4, 6, 7, 8, 9, 11, 13, 14, 15, 18, 21, and 26. The most significant signal was on chromosome 6 (16,934,374 to 17,739,115 bp). In the HW lineage, 34 significant signals (FDR < 5%) were on chromosomes 1, 2, 3, 4, 5, 8, 9, 10, 11, 12, 20, and 24. The two most significant signals were located on chromosomes 2 and 4.

### 3.2. Identifying Purging Regions

Significant signals observed in *hapFLK* analyses may occur either from purging after relaxation of selection or ongoing selection. To distinguish between these two scenarios, *hapFLK* trees were used. The genome-wide *hapFLK* trees show that purging affected a large part of the genome in LW lineage (Figure 2A), while ongoing selection was dominating in HW lineage (Figure 2D). The regional *hapFLK* trees constructed for each region displaying significant haplotype frequency changes (Figure 1) identified eleven of these as purging regions: five in LW and six in HW lineages (Figure 1; Appendix A). In LW, the purging regions were located on chromosomes 1, 6, 7, and 13 (Figure 2B,C; Appendix A), with the strongest signal on chromosome 6 (24.7–25.4 Mb; Figure 2C).

### 3.3. Identifying Candidate Loci Involved in Negatively Correlated Selection Responses for Body Weight and Fitness

We screened for loci involved in negatively correlated responses between 56-day body weight and fitness. This was done by first identifying the major haplotypes in the selected and relaxed lines. Next, testing for associations between these haplotypes and 56-day body weights in the AIL were performed. This with the objective to identify the strongest purging candidates as those where the major haplotypes in the selected and relaxed lines had opposite effects on body weight.

#### 3.3.1. Purging Loci Associated with 56-Day Body Weight in the Low Body Weight Lineage

In the LW lineage, five purging regions containing in total 4,938 SNPs (3,068 with > 0.5 in AFD between the relaxed and selected lines; Appendix A) were evaluated for associations with 56-day body weight in the AIL.

In the proximal region on chromosome 6 (16.9–17.7 Mb), one SNP was significant in the backward-elimination association analysis to body weight at a 20% FDR. The changes in frequency of the more common haplotypes tagged by these SNPs in the selected and relaxed lines were consistent with purging at this locus (Figure 3A). One haplotype (*Hap1_LC6_17_*) increased in frequency from 0.77 in LWS41 to 0.98 in LWS53 and decreased in frequency to 0.16 in LWR9. *Hap2_LC6_17_* was most common in LWR9 (0.84), an increase from LWS41 (0.23). Significant associations with body weight were detected to both haplotypes (*p* < 0.001; ANOVA). Two copies of *Hap1_LC6_17_* resulted in a significantly lower body weight than either no or one copy (Figure 3C; *p* < 0.001; *t*-test), and two copies of *Hap2_LC6_17_* resulted in a significantly higher 56-day body weight (Figure 3B; *p* < 0.001; *t*-test) than either no or one copy.

In the distal region on chromosome 6 (24.7–25.4 Mb), four SNPs were kept at a 20% FDR in the backward-elimination analysis. The changes in frequency of the haplotypes tagged by these SNPs were consistent with purging in the selected and relaxed populations (Appendix A). The haplotype association analyses did not, however, reveal any individually significant effects on 56-day body weight (Appendix A). On chromosome 13, three SNPs were associated with body weight at 20% FDR in the backward-elimination analysis. Significant associations with body weight were observed for the haplotypes constructed from the alleles at these SNPs (Appendix A). However, the changes in the corresponding haplotype frequencies were not consistent with purging in the selected and relaxed lines (Appendix A). In the remaining three purging regions on chromosomes 1 and 7, no SNPs were significantly associated with body weight at 20% FDR in the backward-elimination analysis.

#### 3.3.2. Purging Loci Associated with 56-Day Body Weight in the High Body Weight Lineage

In HWS, six purging regions covering in total 6,558 SNPs (5,342 with > 0.5 in AFD between the relaxed and selected lines; Appendix A) were evaluated for associations with body weight in the AIL.

In the purging region on chromosome 20 (10.7–10.8 Mb), two SNPs were selected at 20% FDR in the backward-elimination analysis. The frequencies of the haplotypes tagged by these associated SNPs across the selected and relaxed lines were consistent with purging (Figure 4A). In the haplotype association analyses, *Hap1_HC20_10_* was significantly associated with 56-day body weight (Figure 4B; *p* < 0.01; ANOVA), as chickens with 2 copies of *Hap1_HC20_10_* had significantly lower 56-day body weights than those with no copies of this haplotype (*p* < 0.05; *t*-test). The most common haplotype (*Hap2_HC20_10_*) in HWS was significantly associated with body weight, as two copies of this haplotype resulted in higher body weight than those without copies of them (*p* < 0.05, Figure 4C). *Hap3_HC20_10_* was also significantly related to body weight, but its frequency in HWS was not consistent with purging (Appendix A).

In the purging region on chromosome 8 (2.4–2.7 Mb), four SNPs were selected at 20% FDR in the backward-elimination analysis. The changes in frequencies of the haplotypes tagged by the associated SNPs across the selected and relaxed lines were consistent with purging in this region (Figure 4D). The most common haplotype in HWS chickens (*Hap1_HC8_2_*) that was lost in the relaxed line (*Hap1_HC8_2_*) was significantly associated with 56-day body weight in the AIL (Figure 4E; *p* < 0.05; ANOVA), as chickens with two copies of this haplotype were heavier than those without any copy of this haplotype (*p* < 0.05; *t*-test). The most common haplotype in the relaxed line (*Hap2_HC8_2_*) was very rare in the HWS lineage as well as the AIL, and there was no association with body weight (Appendix A). The second most common haplotype in the relaxed line (*Hap3_HC8_2_*, frequency = 0.28) was, however, more common in the HW lineage and the AIL where it was significantly associated with 56-day body weight. A single copy of this haplotype resulted in a significantly lower body weight than having no copies of it (*p* < 0.01, Figure 4E).

In the purging region on chromosome 2 (15.9–16.9 Mb), the backward-elimination analysis retained six SNPs at 20% FDR, and in the region on chromosome 4 (35.2–35.2 Mb) it retained one. However, no significant associations were found between the individual haplotypes tagged by the SNPs on chromosome 2 and body weight (See Appendix A). In the region on chromosome 4 where significant associations between the haplotypes tagged by the retained SNPs and body weights were detected, the changes in frequencies of the associated haplotypes were not consistent with purging (Appendix A).

In the purging region on chromosome 9 (11.9–13.6 Mb), 32 SNPs were detected, however, no association between haplotypes tagged by these SNPs and body weight were found (Appendix A). In the purging region on chromosome 11 (10.6–11.3 Mb), six SNPs were found. Although the frequencies of haplotypes tagged by SNPs on chromosome 11 were not consistent with purging in the relaxed populations, they were significantly associated with body weight (Appendix A).

#### 3.3.3. Total Variance in 56-Day Body Weight Explained by the Purging Loci

The total variance in 56-day body weight explained by the eleven purging regions identified in the LW (5) and HW (6) lineages was estimated by fitting the 58 SNPs retained in the backward-elimination analysis across the purging loci in the AIL (Appendix A) using model (1). In total, these SNPs explained 3.4% of the residual phenotypic variance, with 0.3/3.1% being explained of by the 8/50 SNPs selected in the purging regions detected in the LW/HW lineages, respectively.

## 4. Discussion

In a previous study, genome-wide purging was identified after relaxation of selection in the high- and low body weight selected Virginia chicken lines [8]. The earlier study, however, did not identify individual loci contributing to purging, nor did it quantify potential contributions of purging loci to negatively correlated selection responses or fitness. Here, we extended this work in multiple ways. First, by using whole-genome sequence density markers multiple individual purging loci could be identified in both lineages at high resolution. Next, by using genotypic and phenotypic data from a large AIL population, we could evaluate which of the identified purging loci the segregating haplotypes were associated with 56-day body weight. Under the assumption that body weight was a proxy for overall fitness of the chickens in this population, we could identify loci where selected and purged alleles had opposite effects on body weight as likely candidates contributing to negatively correlated selection responses. These fulfill the aims of this study of identifying the individual loci contributing to the purging and highlight those most likely involved in the negatively correlated selection responses on body weight and fitness related traits earlier observed in this population [12,13,15,24].

Throughout the long-term experiment, individual selection has been practiced within the respective HWS and LWS lines, and by avoiding matings among half sib or close relatives, the resulting inbreeding is moderate given the low effective population sizes (N_eHWS_ = 32 and N_eLWS_ = 38, respectively) [13,15,25,26]. Now, after more than 60 generations of bi-directional selection, there is an approximately 15-fold difference in body weight between them [26]. However, the selection responses in body weights were accompanied by correlated responses in traits related to overall fitness in both the LWS and HWS. With overall fitness, we mean the general ability of individuals in the population to survive and reproduce. It therefore involves many different genetic and physiological components altered during the course of the selection experiment, including appetite, immune responses, reproduction, and metabolism [12,13,15,24,26]. We earlier showed that both the HWR and LWR display significant purging on a genome-wide level when compared to their corresponding selected line (HWS and LWS) [8]. It was, however, not explored which individual loci contributed to these signals and consequently not whether the purged alleles were associated with any negatively correlated responses, or decreases in overall fitness, of the chickens in the selected lines.

### 4.1. Genome-Wide Effects of Relaxing Selection in the Low and High Body Weight Selected Lineages

Genome-wide, the purging signal was stronger in the LW than in the HW lineage which is consistent with earlier analyses of these lines [8]. In total, 11 individual purging regions (5/6; LW/HW lineages) were identified. That there were fewer purging loci in the LW lineage may be surprising given its stronger genome-wide purging signal. There are several possible explanations for this. One is that the selection-plateau that was reached in the LWS after about 30 generations has been maintained by balancing selection across multiple loci with deleterious alleles [13]. Their maintenance at intermediate allele-frequencies across generations would lead to weaker signals in the hapFLK analysis [22] as there is no contributing selection-signal at these loci. Another explanation is that selection for lower body weights in the LWS was selection for decreased overall fitness, which is likely a trait with a highly polygenic genetic architecture where many loci contribute small negative effects [27]. Relaxation of selection would then result in an overall increase in fitness by a slower purging of alleles in each individual locus. This would be consistent with a genome-wide purging signal with less pronounced effects at individual loci. It can, given the current data, not be excluded that some of the observed purging signals were due to drift rather than selection, but it may be considered unlikely in those where also associations to body weights were found in the AIL.

### 4.2. A Majority of the Purging Regions Show Associations with Body Weight

One aim of the study was to identify purging loci that potentially contributed to negatively correlated selection responses in the Virginia body weight lines and identifying putative candidate genes for those effects. However, neither direct nor indirect estimations of the putative contributions by these loci to negatively correlated selection responses were possible directly in the selected lines, because they are too small for estimating the phenotypic effects of the involved haplotypes. Such evaluations, however, were possible by screening for associations between the purging loci and 56-day body weights in the AIL (*n* = 2667; F2–F15) developed from LWS and HWS [28]. Body weight was not in itself a direct measure of fitness, however, as when selection for it was relaxed it allowed for natural selection, because (i) it was the target trait for the artificial, directional selection in the LWS/HWS, and (ii) associations between it and haplotypes identified in the purging analysis in the AIL provided additional support for the purging signal being of biological relevance. There are, however, reasons for why associations are not expected at all loci using this approach. First, the power in the association tests is low for selected and purged haplotypes present at low frequencies in LWS/HWS founders for the AIL (Appendix A). The AIL was bred from founders from the selected lines after 40 generations of selection (generation S41) when many body weight loci may be expected to be at or near fixation [8,14]. Thus, for example, their effects are small, new mutations have arisen recently or they are under balancing selection due to negative effects on overall fitness. Smaller effective sizes and lower minor allele frequencies are thus expected to lead to less variance explained by the loci and decreased statistical power in the same way as in GWAS analyses in general populations [29]. Second, it is not unlikely that the haplotypes have different effects on body weight in the AIL, where the genetic and physiological variation is less than in the phenotypically and genetically more extremes in the selected lines. Such interactions (gene x gene and gene x environment) are not possible to disentangle in the current analyses. Third, although body weight was the focal trait in the selected lines, one should not expect all purged haplotypes to have direct effects on it as their major effects may be on other fitness traits. In the Kauai chicken, for example, it was observed that feralization resulted in signatures of reversed selection across the genome, rather than reversing the effects of domestication [30]. Although the SNP selected in the 11 purging regions together only explained 3% of the variance in the AIL, associations with 56-day body weights were observed in 6 of the 11 purging regions at a 20% FDR. Further, in three of these regions significant associations to individual haplotypes were found at *p* < 0.05. Together, they provide additional support for the use of the selected and relaxed Virginia body weight chickens to identify individual loci for further exploration of how individual loci and haplotypes therein contribute to negatively correlated selection responses.

### 4.3. Candidate Genes Located in Purging Regions

For nearly forty years, it has been known that the sense of taste was associated with the systemic control of feed intake in the Virginia body weight lines [31], however, the genetic basis for the distinct responses to same food flavors of the HWS and LWS are still unknown. In light of this, our observation that the strongest purging signal in the LW lineage, on chromosome 6 (Figure 1A; Figure 3A–C), covers the gene, *PKD2L1* was interesting. This gene has been reported as a candidate receptor for sour taste in mammals [32], and it may be hypothesized that mutations in this gustatory gene might contribute to differences in perception of taste. That the haplotype association analysis in this region showed a significant association to 56-day body weight, where the purged haplotype increased body weight relative to the selected haplotype, makes it a relevant candidate for studying its potential involvement in decreased food intake and body weights in the LWS [13].

In the HW lineage, a strong purging signal was observed on chromosome 20 (Figure 1B). The significant association to body weight at this locus was consistent with the effect of the purged haplotype (Figure 4A–C). This region covers a candidate gene, *DNTTIP1*, due to its role as a target for *DLK1* [33] which is the causal gene for muscle hypertrophy in callipyge sheep [34]. Callipyge sheep display a postnatal muscle hypertrophy with 35–40% increased muscle mass and 6–7% decreased carcass fat [35,36], and has decreased meat tenderness [37]. Although imprinting has not yet been observed in birds, it still can be hypothesized that selection on variants at a downstream target of the *Callipyge* locus could influence body weight also in chickens, as well as other domestic animals.

## 5. Conclusions

Using long-term selected, relaxed, and advanced intercross populations from the Virginia body weight chicken lines we detected several purging loci, and estimated the effects on 56-day body weights of the haplotypes in these that were purged after relaxation of selection. Because body weight was the only selected trait in the lines, it was used as an indicator of overall fitness to identify loci and haplotypes likely to be involved in observed negatively correlated selection responses in these lines. Two loci, containing functional candidate genes for appetite and muscle development, were identified as top candidates for subsequent functional genetic studies. Together, our results illustrate the value of long-term experimental selection experiments for the empirical detection of haplotype purging and mapping of loci that likely contribute to correlated responses under artificial selection. The identified loci and candidate genes are interesting targets for future functional genetic research to identify the molecular mechanisms underlying correlated selection responses in domestic animals.

## Figures and Tables

**Figure 1 genes-11-00630-f001:**
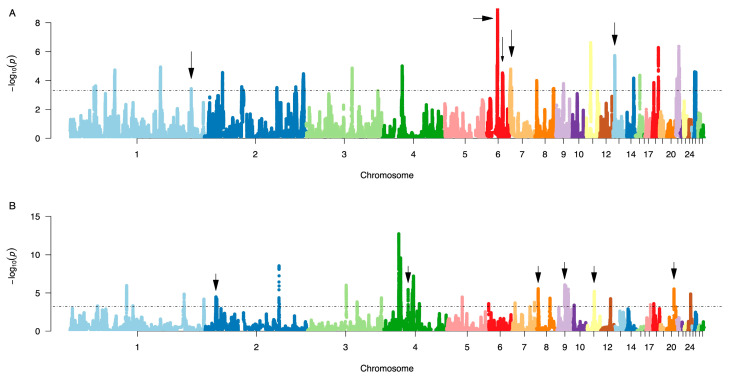
Genome-wide scans for regions with significant haplotype changes. (**A**) LW and (**B**) HW selected Virginia chicken lineages. Purging regions (see 3.2. below) are highlighted by black arrows.

**Figure 2 genes-11-00630-f002:**
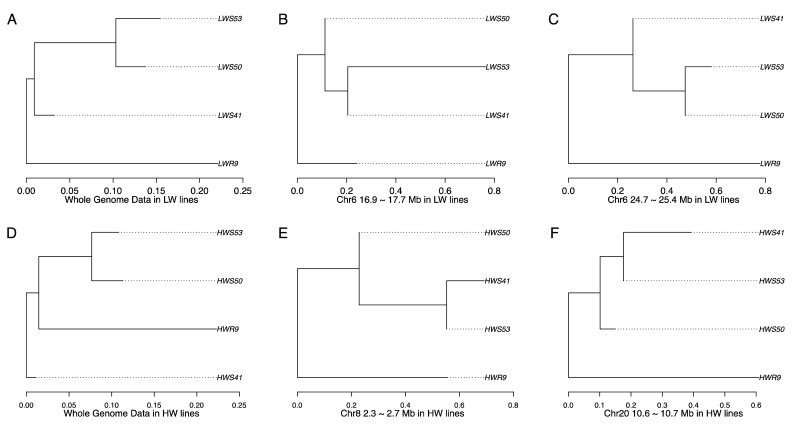
Genome-wide and regional *hapFLK* trees identifying purging in the HW and LW lineages. Panels (**A**–**C**) show trees for the LW lineage, with (**A**) suggesting strong overall genome-wide purging and (**B**,**C**) illustrating the two strongest regional purging signals on chromosome 6. Panels (**D**,**F**) show the *hapFLK* trees for the HW lineage, with (**D**) illustrating that ongoing selection dominated on the genome-wide level and (**E**,**F**) illustrating the strongest purging signals on chromosomes 8 and 20. The trees for the remaining seven purging regions are provided in Appendix A.

**Figure 3 genes-11-00630-f003:**
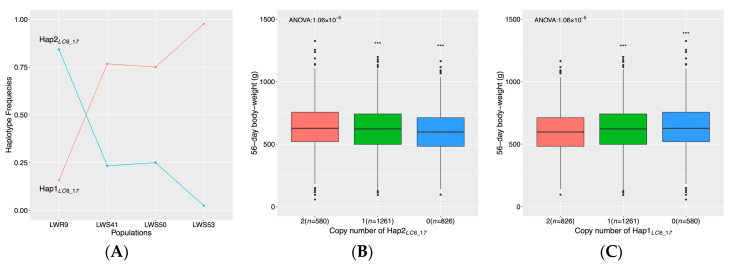
Purging regions in the LW lineage with associations to 56-day body weight. The figure presents results for the candidate purging region around 17Mb on chromosome 6. In (**A**), the large changes in frequencies consistent with purging were observed for haplotypes *Hap1_LC6_17_* and *Hap2_LC6_17_* across the four LW populations. In (**B**,**C**), box plots of 56-day body weights are shown for chickens with no, one, and two copies of these haplotypes (Purged *Hap2_LC6_17_* in (**B**); Selected *Hap1_LC6_17_* in (**C**). Overall *p*-values in (**B**)/(**C**) were from the one-way ANOVA analysis. Stars denote the significance of pair-wise *t*-tests between haplo-genotypes (***/*/ns represent *p* < 0.001/*p* < 0.05/non-significant).

**Figure 4 genes-11-00630-f004:**
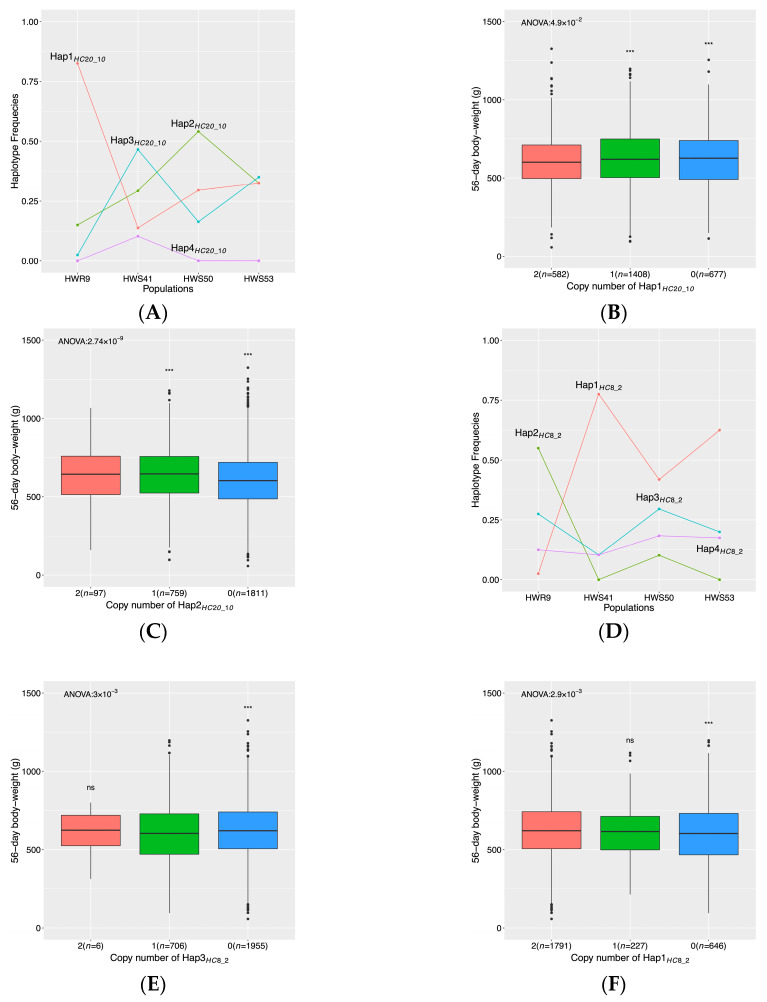
Purging regions in the HW lineage with associations to 56-day body weight. The top panel presents results for the purging region on chromosome 20 (10.7–10.8 Mb). In (**A**), the large changes in haplotype frequencies for *Hap1_HC20_10_* and *Hap2_HC20_10_* across the four HW populations are consistent with purging. In (**B**,**C**), box plots of 56-day body weights are shown for chickens with no, one, and two copies of these haplotypes (*Hap1_HC20_10_* in (**B**); *Hap2_HC20_10_* in (**C**)). In bottom panel, (**D**) haplotype frequency changes consistent with purging are shown in the purging region on chromosome 8 (2.4–2.7 Mb). In (**E**,**F**), box plots illustrate 56-day body weight distributions for AIL individuals with no, one, and two copies of these haplotypes (*Hap3_HC8_2_* in (**B**); *Hap1_HC8_2_* in (**C**)). Overall *p*-values in B/C/E/F were from the one-way ANOVA analysis. Stars denote the significance of pair-wise tests between genotypes calculated by a *t*-test (*/**/ns represent *p* < 0.05/0.01/non-significant).

**Table 1 genes-11-00630-t001:** 56-day body weights of the Virginia high weight (HW) and low weight (LW) chicken lines (S—selected, R—relaxed). Reported are generation means (grams ± standard deviations) for males and for females. Presented are data from the founder population (generation S0) and for the generations of the HW and LW lines included in this study (generations S41, S50 and S53/R9).

Year	Generation	Base Population			
		**Male**		**Female**	
1957	S_0_	878 ± 123		708 ± 117	
		**HW line**			
		**HWS**		**HWR**	
		Male	Female	Male	Female
1998	S_41_	1684 ± 142	1374 ± 106		
2007	S_50_	1811 ± 179	1501 ± 99		
2010	S_53_/R_9_	1712 ± 218	1381 ± 160	1456 ± 240	1211 ± 163
		**LW line**			
		**LWS**		**LWR**	
		Male	Female	Male	Female
1998	S_41_	241 ± 73	175 ± 42		
2007	S_50_	165 ± 60	117 ± 38		
2010	S_53_/R_9_	134 ± 35	114 ± 41	204 ± 44	156 ± 44

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
