# Peer review of "Haplotype Purging after Relaxation of Selection in Lines of Chickens That Had Undergone Long-Term Selection for High and Low Body Weight"

_genes, 2020, doi:10.3390/genes11060630_

Round 1

Reviewer 1 Report

genes-797537 Haplotype purging after relaxation of selection in lines of chickens that had undergone long-term selection for high and low body weight

This study has identified a few genome regions with haplotype changes associated with bi-directionally selected for body weights in 56-day Virginia chicken population.

The manuscript is interesting and could deliver new information about potential genome regions involved in the determination of body weights in day 56 of life. The investigation was performed included a few chicken generation what is interesting and better show the haplotype changes throughout the selection. I have only minor comments and suggested this manuscript for publication after consideration them.

Minor comments:

  1. Please attach information about Virginia chicken phenotype throughout the investigated generations for both high and low lines body weight in day 56 of life. Because the readers do not know this population the phenotype changes during these generations would illuminate a more complete picture of these populations.
  2. Why the authors used Gallus_gallus – 5.0 and not the newest 6.0 as a reference?
  3. Discussion section – line 318-365 here is very few citations and this part looks like the results, not the discussion, could authors add more references for this part
  4. The discussion should include a few information about Virginia lines, about the start bodyweight in 56-day (before selection) and what was achieved after these numerous generations both in low and high lines.
  5. Line 379-382 the locus callipyge is paternal imprinted in mammals and the muscular hypertrophy is strongly associated with imprinting phenome. And in birds the imprinting was not observed. Add one or two sentences of comments.
  6. The numbers in supplementary files should be aligned the same number of decimal places
  7. Why SNPs in tables S2 even far located have the same frequency? it is associated with imputation? please comment in the footnote
  8. Did the Virginia chicken populations (high and low) were selected before for any SNP?
  9. Did all SNPs present in the supplementary tables were included in the model?
  10. Why in the model were used only two fix effects (sex and generation), why not hatching effect or interaction between factors were not included, what about the relationship between investigated chickens, they are after different hens, what about the roosters?

Author Response

Reviewer 1

genes-797537 Haplotype purging after relaxation of selection in lines of chickens that had undergone long-term selection for high and low body weight

This study has identified a few genome regions with haplotype changes associated with bi-directionally selected for body weights in 56-day Virginia chicken population.

The manuscript is interesting and could deliver new information about potential genome regions involved in the determination of body weights in day 56 of life. The investigation was performed included a few chicken generation what is interesting and better show the haplotype changes throughout the selection.  I have only minor comments and suggested this manuscript for publication after consideration them.

Minor comments:

Q1: Please attach information about Virginia chicken phenotype throughout the investigated generations for both high and low lines body weight in day 56 of life. Because the readers do not know this population the phenotype changes during these generations would illuminate a more complete picture of these populations.

R1: A table with 56-day body-weights for the founder population and investigated generations of the selected and relaxed lines has now been included in the revised manuscript.

Q2: Why the authors used Gallus_gallus – 5.0 and not the newest 6.0 as a reference?

R2: The re-sequencing data used in this study were produced as part of earlier studies and were then processed using the older GG5 reference. As the work, and computational resources, required for remapping these data to newest reference genome are considerable,  we decided, based on an evaluation of the differences between the reference genomes v5.0 and v6.0, that the changes between the assemblies were so small on the autosomes (which were the ones targeted in this study) that they were only likely to have a minor impact on the results in this case. Readers interested in comparison with later studies based on GG6 reference would have the opportunities to lift positions from GG5 over to GG6. Therefore, it was decided that the additional work required to do the remapping of the data were not essential.

Q3: Discussion section – line 318-365 here is very few citations and this part looks like the results, not the discussion, could authors add more references for this part

R3: These two sections in the discussion have now been shortened by removing results and focusing more on discussion. Relevant references have also been included. We hope that these changes have addressed this comment.

Q4: The discussion should include a few information about Virginia lines, about the start bodyweight in 56-day (before selection) and what was achieved after these numerous generations both in low and high lines.

R4: In light of comments by this and the other reviewer, the background and development of the Virginia lines pedigree has now been included i) an expanded material and methods and iii) in the form of highlighting the references that cover the selected, relaxed and AIL populations in greater detail in the discussion. Further, a new table (see R1 above) with 56-day BW was added to demonstrate the BW changes during the experiment. We hope that these changes address this question.

Q5: Line 379-382 the locus callipyge is paternal imprinted in mammals and the muscular hypertrophy is strongly associated with imprinting phenome. And in birds the imprinting was not observed. Add one or two sentences of comments.

R5: We have now included a sentence clarifying that imprinting has not yet been shown in birds and how it relates to this locus.

Q6: The numbers in supplementary files should be aligned the same number of decimal places

R6: We have now updated the supplementary files such that all numbers are given with the same number of decimal places.

Q7: Why SNPs in tables S2 even far located have the same frequency? it is associated with imputation? please comment in the footnote

R7: We assume that the referee here refers to all Supplementary Tables S2-S11 that contain this type of information. First, we want to note that the number reported is not the allele-frequency but instead the allele-frequency difference between the populations in these regions. Next, we expect to observe this result for two reasons. They are i) all markers that tag the same haplotypes in these regions will result in the same allele-frequency difference in the table, and ii) that the regions, although they might seem long due to their containing many markers, are still rather short in kbp and given the short time of divergence between populations haplotype-sharing across such distances are still likely for some regions.

Q8: Did the Virginia chicken populations (high and low) were selected before for any SNP?

R8: No, the same selection regime has been used since 1957 when the experiment was started (pre-SNP era) and has been for phenotype only.

Q9: Did all SNPs present in the supplementary tables were included in the model?

R9: All the SNPs reported in the supplementary tables have been tested for association to the phenotype using the described backward-elimination procedure. However, simultaneously fitting all ~ 970  SNPs in a multi-locus linear model was not possible due to over-parameterization Therefore, several rounds of analysis, each with ~200 markers evaluated, were performed to select the final set of markers. Markers that are highlighted with green in the supplementary tables are those that remained at the desired FDR threshold. These were subsequently used to tag the haplotypes, and tested for associations between these haplotypes and 56-day body weights in the Advanced Intercross Line.

Q10: Why in the model were used only two fix effects (sex and generation), why not hatching effect or interaction between factors were not included,

R10: The model used in the association analyses was chosen based on earlier experiences from analysing data from generations F2-F8 of this AIL (Besnier et al. Fine mapping and replication of QTL in outbred chicken advanced intercross lines. Genet Sel Evol 2011 43:3. doi: 10.1186/1297-9686-43-3), which contributed the majority of the samples used here. In each generation, the chicks were produced from a single hatch on the first Tuesday of every March. Therefore, the hatch effects were homogeneous within generation and completely confounded with generation effects across generation.  In the analysis,  we only included generation as a factor to capture both these effects. The additional effects explored, but ultimately not included in the model, were those of genetic background tested by including markers from earlier mapped regions (Zan et al Mol Biol Evol. (2017) 34:2678-2689. doi: 10.1093/molbev/msx194.). They were not included as they did not affect the results for the regions reported here. However, given the limited impact of including/excluding other environmental factors in the model on the genetic associations in this population in general, we would not expect this to affect the final results more than marginally.

Q11: what about the relationship between investigated chickens, they are after different hens, what about the roosters?

R11: The breeding of the HWS and LWS are now described in more detail in the manuscript and previous publications, starting with reference #19. We hope that the revisions made have addressed this point.

Reviewer 2 Report

Major comments

The manuscript has some interesting findings. The analysis sounds appropriate. However, the manuscript is extremely poorly written and needs to be largely reworked. Apart from English, which the author should consider language-editing service, the manuscript is not written in a proper academic manner. The introduction does not show the rationale of your study. The objectives are not clear. The discussion does not explain your findings. The conclusion is unclear and irrelevant. From your introduction and conclusion, to be honest, I can’t see any benefits or contributions of this paper to the current knowledge although some findings may be good.

The authors mentioned about fitness although no actual fitness traits were measured. After reading the whole paper, I realize that you studied/implied fitness through investigating BW of lines. This is quite interesting, but “hidden too well”. How you define or study fitness should be clearly introduced in the introduction. For example, fitness is hard-to-measure traits.

Specific comments

Abstract

Lines 14-20: Too long introduction, but without clear objectives of the study. I can’t see the importance of this study from your abstract.

Lines 21-24: I am not sure these are your objectives or methods

Line 27: Please don’t use the word ‘novel’.

Lines 31-32: bad conclusions.

Lines 33-35: delete this sentence. You can mention this in discussion, but don’t waste valuable word counts here. It adds nothing.

Introduction

You should describe a bit about the two lines, how they were selected (selection methods), previous studies (I can find number of studies related to these two lines). Are there any similar studies? What is missing elements from the previous findings. Define fitness…

I can not find the objectives of this study.

Is it necessary for the use of both LWS and LW lineage? Or HWS and HW?

Lines 40-57: totally irrelevant. Please delete this paragraph.

Lines 53, 294: “competition in the allocation of resources”; “Under artificial or natural selection, the resources available to an individual are finite”.

I strongly disagree with these ideas in number of ways. Or you put in the wrong context. Did LWS have better fitness than HWS? And how much better? 15-times better?

We can increase birds’ capability of ingesting more resources, can’t we?

Note that selection may have negative effects on other traits, I agree.

Lines 73-82: are these results, discussions and conclusions? I think you need to go through academic writing classes.

Materials and Methods

A very important information that is missing in M&M is how animals are selected for mating. Phenotypic selection, pedigree-based BLUP, or genomic-based BLUP? What is relaxed selection?

A descriptive statistic table on BW would be nice, but not necessary. It is up to you of course.

Lines 84-87: is this ethics statement so essential that you have to put it first? You only analyzed data, didn’t you? You should focus on the more important parts of the method. Anyway, I am not sure about rules of this journal.

Lines 165, 174 Equation (1) & (2): are these models in scalar or matrix notations? If it is scalar, it is not written correctly.

Results

Lines 207-211: This looks like M&M to me.

Discussion

Lines 293-311: This looks like introductions and recommendations to me.

Line 319: I don’t see “greater detail” here. By the way, is your study better (in term of accuracy of identifying purging region) than previous study? And why it’s better? And why your and their findings are different? I think these should be detailed.

Is this possible the purging regions due to drift? What are the effective population size and breeding structures for these lines?

Line 327: do you have data on “overall fitness”, for example mortality rate?

Conclusion

Lines 382-385: This is introduction? Citation in conclusion?

Lines 393-396: This goes to discussion.

We need a real conclusion as a take-home message, here. Please re-formulate.

Author Response

Reviewer 2:

Major comments

The manuscript has some interesting findings. The analysis sounds appropriate.

Q1:However, the manuscript is extremely poorly written and needs to be largely reworked.

R1: The manuscript has now been extensively revised and we hope that by the redesign of the manuscript, the previously hidden parts are now clearer.

Q2: Apart from English, which the author should consider language-editing service,

R2: We have revised the manuscript extensively and hope the revised manuscript now meets the standard for publication.

Q3: the manuscript is not written in a proper academic manner. The introduction does not show the rationale of your study. The objectives are not clear. The discussion does not explain your findings. The conclusion is unclear and irrelevant. From your introduction and conclusion, to be honest, I can’t see any benefits or contributions of this paper to the current knowledge although some findings may be good.

R3: In the revisions, changes were made as suggested by this reviewer to improve the introduction, clarify the objectives, better explain the findings in the discussion and clarify the conclusion by making it more relevant.

Q4: The authors mentioned about fitness although no actual fitness traits were measured. After reading the whole paper, I realize that you studied/implied fitness through investigating BW of lines. This is quite interesting, but “hidden too well”. How you define or study fitness should be clearly introduced in the introduction. For example, fitness is hard-to-measure traits.

R4: The introduction has now been thoroughly revised to also include a description of how we define, and attempt to study fitness indirectly via BW’s, in more detail. We hope that this has clarified this point.

Specific comments

Abstract

Q5: Lines 14-20: Too long introduction, but without clear objectives of the study. I can’t see the importance of this study from your abstract.

R5: The abstract has now been modified and rewritten to more clearly specify the objectives and importance of the study. We hope these revisions address this point.

Q6: Lines 21-24: I am not sure these are your objectives or methods

R6: We hope that this has been clarified in by the revisions to this section.

Q7: Line 27: Please don’t use the word ‘novel’.

R7: The word ‘novel’ has been removed.

Q8: Lines 31-32: bad conclusions.

R8: We hope that the changes made have provided clarification.

Q9: Lines 33-35: delete this sentence. You can mention this in discussion, but don’t waste valuable word counts here. It adds nothing.

R9: We have followed the advice and removed the sentence.

Introduction

Q10: You should describe a bit about the two lines, how they were selected (selection methods), previous studies (I can find number of studies related to these two lines). Are there any similar studies? What is missing elements from the previous findings. Define fitness…

R10: The Virginia lines are now more thoroughly introduced in the Introduction of the revised paper and described in more detail in the Materials and Methods section (see also response to Q1 from Reviewer 1). References to work describing the development of the lines are also given in both of these places. Further, connections to similar earlier work are provided and their missing elements highlighted. Also our definition and use of fitness has been added to the introduction (see also R4 above). We hope these revisions provide clarification.

Q11: I can not find the objectives of this study.

R11: We have now defined the objectives of the study more clearly both in the abstract (see R6 above) and in the Introduction.

Q12: Is it necessary for the use of both LWS and LW lineage? Or HWS and HW?

R12: Yes, because ‘XWS’ refers to the body-weight selected lines specifically, whereas the “XW-lineage” also includes the respective relaxed lines.

Q13: Lines 40-57: totally irrelevant. Please delete this paragraph.

R13: This section was included to connect the study to the topic of the special issue of the journal. We have for this reason kept it in the revision, but are open to removing it based on editorial advice.

Q14: Lines 53, 294: “competition in the allocation of resources”; “Under artificial or natural selection, the resources available to an individual are finite”. I strongly disagree with these ideas in number of ways. Or you put in the wrong context. Did LWS have better fitness than HWS? And how much better? 15-times better?

R14: In the selection experiments, a reduction in fitness was observed in both the HWS and LWS lines (detailed information on the lines/fitness etc are covered in detail beyond what is possible here in Dunnington et al (1996 & 2013), Márquez et al (2010), Siegel et al (1966), Farielle et al (2013), Jambui et al (2017) included as references 9,10,12-15 in the manuscript. Relaxing selection allows natural selection to come into play, but it is not possible to return to origin because there is, for example, loss of genetic material both regarding additively acting alleles and combinations of alleles across multiple loci. The statement in the discussion section (l294) has been removed during the revision of the discussion. We hope that the statement, in the new context of the revised introduction and the responses above provide clarification.

Q15: We can increase birds’ capability of ingesting more resources, can’t we?

R15: The point made here was not intended to relate only to the ingestion of more resources, but also differences in allocations during different stages (growth, reproduction etc) in life. We hope that the revisions to the manuscriptprovide clarification.

Q16: Note that selection may have negative effects on other traits, I agree.

R16: We appreciate hearing that.

Q17: Lines 73-82: are these results, discussions and conclusions? I think you need to go through academic writing classes.

R17: The aim was to briefly highlight the contributions of the study and this part of the introduction has now been extensively revised.

Materials and Methods 

Q18: A very important information that is missing in M&M is how animals are selected for mating. Phenotypic selection, pedigree-based BLUP, or genomic-based BLUP? What is relaxed selection?

R18: The Materials and Methods section has now been significantly revised to describe how the HWS and LWS, and their corresponding relaxed lines, were selected and bred. See also response R11 to reviewer 1.

Q19: A descriptive statistic table on BW would be nice, but not necessary. It is up to you of course.

R19: A table (Table 1) providing this information is now, as requested, included in the revised manuscript (see also R1, Reviewer 1).

Q20: Lines 84-87: is this ethics statement so essential that you have to put it first? You only analyzed data, didn’t you? You should focus on the more important parts of the method. Anyway, I am not sure about rules of this journal.

R20: We included this to be entirely open with the ethics regarding the experiments to breed the animals used. We consider this essential because the chickens were raised, blood was drawn from them for this study, etc. We are, however, open to locate the statement elsewhere based on journal and/or editorial policy.

Q21: Lines 165, 174 Equation (1) & (2): are these models in scalar or matrix notations? If it is scalar, it is not written correctly.

R21: The model has now been revised to be in scalar notation. We hope they are now written correctly.

Results

Q22: Lines 207-211: This looks like M&M to me.

R22: This section has been removed.

Discussion

Q23: Lines 293-311: This looks like introductions and recommendations to me.

R23: This text has now been partially removed and partially integrated in the extensively revised introduction.

Q24: Line 319: I don’t see “greater detail” here. By the way, is your study better (in term of accuracy of identifying purging region) than previous study? And why it’s better? And why your and their findings are different? I think these should be detailed.

R24: The text has now been reformulated to better explain the new contributions by this study. These include i) improved resolution over of earlier works by using WGS-imputed SNP data, ii) the identification of sweeps accounting for haplotype data, and iii) by identifying individual loci and associating purged haplotypes with 56-day body weight in the AIL.

Q25: Is this possible the purging regions due to drift? What are the effective population size and breeding structures for these lines?

R25: In the revised manuscript, we now acknowledge the possibility that some purging regions might be due to drift, while highlighting that those where associations to body weight were detected are more likely to contain true selected/purged variants. The realized effective population sizes in the HWS/LWS lines were 32.1/38.3, respectively,and 30 for the LWR/HWR. The breeding structures are now more thoroughly described (see R18 above).

Q26: Line 327: do you have data on “overall fitness”, for example mortality rate?

R26: Although it would be valuable to have, the rate at which the chickens died is not a sole measure of fitness. For example, in the low line, some chickens die from being anorexic (never starting to eat), and others will not enter lay unless force fed. In the high line, feeding is restricted after selection age to control for metabolic disorders that increase mortality in these lines. Thus, we use fitness in a generic sense rather than delving into specific fitness components other than mentioning behavior and reproduction.

Conclusion

Q27: Lines 382-385: This is introduction? Citation in conclusion?

R27: The conclusion section has been significantly revised to focus on summarizing/concluding the work. The revised version contains no citations. We hope this addresses this point.

Q28: Lines 393-396: This goes to discussion.

R28: This text has been removed from the conclusions.

Q29: We need a real conclusion as a take-home message, here. Please re-formulate.

R29: The conclusions have now been rewritten and hopefully provide a more specific take-home message.

Round 2

Reviewer 2 Report

Most questions have been responded properly. No further comment.

Author Response

We thank you for the constructive comments that helped to improve the initial manuscript.